# Recent Progress in Smart Electronic Nose Technologies Enabled with Machine Learning Methods

**DOI:** 10.3390/s21227620

**Published:** 2021-11-16

**Authors:** Zhenyi Ye, Yuan Liu, Qiliang Li

**Affiliations:** 1Department of Electrical and Computer Engineering, George Mason University, Fairfax, VA 22030, USA; zye@gmu.edu; 2Applied Materials, Sunnyvale, CA 94085, USA; yuanliu19@gmail.com

**Keywords:** electronic nose, gas sensor array, machine learning, neural networks, review

## Abstract

Machine learning methods enable the electronic nose (E-Nose) for precise odor identification with both qualitative and quantitative analysis. Advanced machine learning methods are crucial for the E-Nose to gain high performance and strengthen its capability in many applications, including robotics, food engineering, environment monitoring, and medical diagnosis. Recently, many machine learning techniques have been studied, developed, and integrated into feature extraction, modeling, and gas sensor drift compensation. The purpose of feature extraction is to keep robust pattern information in raw signals while removing redundancy and noise. With the extracted feature, a proper modeling method can effectively use the information for prediction. In addition, drift compensation is adopted to relieve the model accuracy degradation due to the gas sensor drifting. These recent advances have significantly promoted the prediction accuracy and stability of the E-Nose. This review is engaged to provide a summary of recent progress in advanced machine learning methods in E-Nose technologies and give an insight into new research directions in feature extraction, modeling, and sensor drift compensation.

## 1. Introduction

An electronic nose (or E-Nose) is an aroma analyzer that uses mechanical and electronic components to emulate the human olfactory system. Unlike conventional aroma analysis methods under the lab environment, E-Nose is developed for applications demanding quick measurement while avoiding the subjectivity of humans, which has been proven promising in robotics [1,2,3,4], food engineering [5,6,7,8,9,10,11], environment monitoring [12,13,14,15], and diagnosis of diseases [16,17,18,19,20,21].

Compared to the human olfactory system, an E-Nose uses a gas sensor array to convert the gas molecular signals into electric signals (Figure 1). Although no highly specific receptors are used in an E-Nose, unique patterns can be generated for various odors as their fingerprints for future predictions through proper machine learning techniques. According to Yan et al. [22], most optimizations adopted by recent studies for E-Nose systems belong to one of three categories: sensitive material selection and sensor array optimization, the feature extraction and selection method, and the pattern recognition method. Despite the advancements in finding more selective and sensitive materials/mechanisms for gas sensing such as a functionalized graphene [15,23,24,25,26], the conductive polymer [27,28,29,30], and sound acoustic wave gas sensor [31,32], improving the differentiation capability and long-term signal consistency of an E-Nose remains a challenge for machine learning and data processing.

The general machine learning framework of the E-Nose for specific applications involves feature extraction, modeling, and drift compensation. An E-Nose produces high-dimensional time-series raw signals in responding to target gases, which contain noises and redundant information. Feature extraction preserves only the information uniquely characterizing the pattern of an odor signal. The extracted features can be used for qualitative and quantitative aroma analysis assisted by proper modeling techniques. Qualitative aroma analysis aims to distinguish different odors; quantitative aroma analysis predicts a particular property associated with target odor. However, many gas sensors suffer from drifting problems [34,35,36]; that is, sensor responses to the same gas change over time due to sensor aging and environment change, and the inconsistency of sensor responses may void the model built on previous data. The sensor drifting problem can be relieved by drift compensation algorithms through machine learning instead of building a model on new data.

Although there have been a few previous reviews on the E-Nose, they focus either on specific applications [37,38,39] or only a portion of the entire E-Nose data processing pipeline [22,40]. In comparison, this review aims to provide a comprehensive study of machine learning techniques for general E-Nose applications. Moreover, recent years have seen high-performance neural network approaches adopted by various machine learning tasks for audio and image processing, but few works review their effectiveness in E-Nose data processing. Therefore, we are motivated to include the latest practices that have applied neural networks to the E-Nose in this work. Here, we review the recent advances in E-Nose machine learning techniques, with a focus on three important aspects: (1) feature extraction, (2) modeling, and (3) gas sensor drift compensation. By surveying machine learning methods for different E-Nose applications, this work tries to evaluate the performance of an E-Nose in existing applications and to inspire new applications in many emerging fields.

## 2. Machine Learning for E-Nose

### 2.1. Feature Extraction

E-Nose data is a time-series array of high dimensionality that reflects the concentration of target gas, and the sensor response shows as in Figure 2. The signal strength increases during the response phase and decreases during the sensor recovery phase in each measurement. Odors are distinguished and identified based on their distinct features at both phases in their responses. The features can either be manually extracted or learned from a neural network.

#### 2.1.1. Manual Feature Extraction

In general, manually extracted features are selected based on the prior knowledge of data processing and E-Nose data. Features extracted from raw signals can be either from the time domain or frequency domain [41,42,43].

Time-domain features can be extracted from the original response curve. The commonly used features are summarized by [22] in Table 1, where *x*(*t*) represents either the voltage or resistance change signals generated by the E-Nose. Those features characterize the local pattern and can be calculated based on a small section of complete signals.

Nallon et al. [23] fabricated a graphene gas sensor and tested its response towards 11 different analytes. The signals were first normalized to the range 0~1, after which the undercurve areas during the sensor response and recovery were used as features. Zhi et al. [5] used a commercialized E-Nose with 18 metal oxide semiconductor (MOX) gas sensors to distinguish tea of different qualities. Maximum response values and average response values during the time period were extracted for classification.

In addition to the low-level features abstracting local characteristics of E-Nose signals, there are also high-level feature extraction practices such as parametric fitting with predefined functions [29,30]. Nallon et al. [24] modeled the resistance response of a graphene gas sensor as $Rs\left(t\right)=\alpha_{S}\left(1-e^{-\beta_{s}t}\right)+\gamma_{s}$ for the sensing period and $Rr\left(t\right)=\alpha_{r}e^{-\beta_{r}t}+\gamma_{r}$ for the recovery period. Thus, six function parameters were obtained for each sensor and used as features for gas discrimination. It was reported that those parametric features showed better discrimination towards gases compared to other time-domain features. Yan et al. [44] compared different curve fitting functions on their performance in wound pathogen detection. In the study, signals were collected on seven pathogen samples using an E-Nose device with 5 MOX gas sensors. After the features were extracted using different feature extraction methods, they were used for training the same radial basis function network (RBFN), and the resulting test accuracies were compared. The selection of the parametric fitting function was essential to classification; a template function with more parameters might be influenced by the noise in the signal and result in low-quality features.

Inspired by the parametric curve fitting method, Liu et al. [45] came up with a non-parametric modeling-based feature extraction method by decomposing a sensor signal into impulse responses. An arbitrary signal $v\left(t\right)$ can be represented by $v\left(t|\theta_{s}\right)=\sum_{\tau =1}^{D}\theta_{s}\left(\tau \right)u_{s}\left(t-\tau \right)$, where $u_{s}$ are the ideal step inputs and $\theta_{s}$ are the coefficients used as features. A Mercer kernel was applied to regularize the solution and improve the finite impulse response model, which converted the problem into solving $\hat{\theta_{s}}=argmin_{\theta_{s}}{\left(V_{s}-X_{s}\theta_{s}\right)}^{T}\left(V_{s}-X_{s}\theta_{s}\right)+\theta_{s}^{T}K^{-1}\theta_{s}$, where $V_{s}$ are the sensor response vectors, $X_{s}$ are the vector presentation of $u_{s}\left(t-\tau \right)$, and K is the kernel matrix. The method generated a feature matrix of size D*S, where *D* was the order of impulse response and *S* was the number of sensors. The extracted features were reported as effective for odor classification and noise resistance enough to skip denoising preprocessing.

The Windowing method slices signals in the time domain with window functions such as hamming and gaussian. Guo et al. [46] used a moving window to compute the area surrounded by the window and sensor signals (Figure 3). A window was placed around the peak of the normalized signal and moved both left and right by the width of the window, which generated three features corresponding to each window position. Various hyperparameters were tested including the width and the type of window, of which a training window of size 480 s produced the best classification accuracy.

Frequency-domain features can be extracted after transforming the original time-domain signal to a frequency domain through Fourier transform or wavelet transform [47,48,49,50,51]. During the experiment by Dai et al. [6], who used MOX sensors to classify different teas, original signals were transformed into a vector by wavelet packet decomposition with Daubechies wavelet as the wavelet base. The three-level decomposed signals produced eight energy strengths corresponding to distinct frequency bands for each sensor. Average and maximum energy strengths were then calculated as features. Men et al. [52] applied the same wavelet decomposition method as Dai while using variable importance of projection scores generated by partial least squares (PLS) to choose the features with the most explanatory power. The average and maximum energy strengths were then calculated as features.

Ye et al. [53] transformed the original signal to the frequency domain using discrete Fourier transform and used AC noise shift as a differentiation indicator. The AC noise that came from the power supply of the measurement equipment were 60 Hz and 120 Hz and they would be shifted from the measurement due to the surface reaction of the molecule and gas sensor. Similarly, Gomri et al. [54] analyzed the power spectral density (PSD) of the noise measured at gas sensor terminals during the sensor interaction with target analytes. Features including the derivative and the max power of noise PSD were proved effectively discriminate two different gases.

#### 2.1.2. Feature Extraction through Learning

In addition to manual feature extraction that requires domain knowledge about the E-Nose and gas sensors, features can also be “learned” through optimizing a neural network. Different from the convolution neural network (CNN) in the next section, the methods discussed in this section do not adopt an end-to-end (E2E) scheme, which means they are not trained for prediction. Instead, these methods involve learning intermediate feature representations of E-Nose signals via supervised and unsupervised learning. Shi et al. [55] developed an odor classification pipeline with a CNN and support vector machine (SVM). The CNN was pretrained to learn feature embeddings, after which the SVM was trained on the features for odor classification. It was reported that the workflow cascading the CNN with the SVM had better differentiation ability compared to training two classifiers individually. With a similar architecture, [56] used a CNN to extract fusion features from an E-Nose and hyperspectral data for rice classification. Signals collected from the E-Nose and hyperspectral imager were concatenated and reshaped into a 2D image, after which the CNN was pretrained on the images for feature extraction. Another extreme learning machine (ELM) classifier was then trained on the resulting features for prediction.
(1)$$P\left(v_{j}|h\right)=\sigma \left(c_{i}+\underset{j}{\sum }W_{ij}h_{j}\right)$$
(2)$$P\left(h_{j}|v\right)=\sigma \left(b_{j}+\underset{i}{\sum }W_{ij}h_{j}\right)$$

Langkvist and Loutfi [57] proposed a method that used a Restricted Boltzmann Machine (RBM) to extract features from the E-Nose signals. RBM is a generative model learning to reconstruct input from its hidden representation. More specifically, the researchers used a variation of RBM called conditional RBM (or cRBM) to learn the feature representation of time-series gas sensor array signals (Figure 4). As an unsupervised learning method, RBM does not require labeled data during training, the goal of training is to construct the conditional probability pair as Equations (1) and (2), where *h* is the latent representation of the input, and *v* is the original input. After pretraining cRBM with data, a weight projecting input to latent features can be used to train or fine-tune another model to perform classification.

Other unsupervised models were also tried for feature extraction such as autoencoder and a deep belief network [34,57,58,59]. Autoencoder is a feed-forward network structure aimed at recovering input at its output by minimizing mean square error; a stacked autoencoder refers to an autoencoder with more than one hidden layer. Essentially, an autoencoder is a non-linear version of principal component analysis (PCA) whose non-linearity is introduced by the activation function between hidden layers, and both PCA and autoencoder can be used for dimension reduction [60]. Compared to dimension reduction with PCA, autoencoder can preserve more non-linear relationships in the resulting feature space. A recent study [61] also showed that an autoencoder network built from unlabeled data can generate highly discriminative features for another labeled dataset. Zhao et al. [62] proposed a stacked sparse autoencoder model (SSAE), which was combined with a backpropagation neural network (BPNN) to perform feature extraction for Chinese liquor classification (Figure 5). After the model was trained, an extra prediction layer was appended to the encoder of autoencoder for prediction. Lu et al. [63] replaced the hand-craft features for the E-Nose with latent representation generated from a gated recurrent unit-based autoencoder (GRU-AE). Compared to other dimensionality reduction methods including PCA and Kernel-PCA, feature representations from the GRU-AE were more distinguishable and effectively improved classification performance.

There have been many time-series unsupervised feature extraction methods proposed but not yet tried with gas sensor array signals. A more advanced model involves a temporal autoencoder [64] to capture both short-term features using CNN and temporal changes using long short-term memory (LSTM), or directly applying an LSTM autoencoder to extract features [65].

### 2.2. Modeling

Machine learning models have been heavily researched for mapping E-Nose features to target predictions such as odor categories and gas mixtures of different chemical concentrations.

#### 2.2.1. Qualitative Aroma Analysis

The qualitative aroma analysis with an E-Nose aims to identify the distinctness of several unknown gas samples by the responses generated from the E-Nose device. The diversity of gas sensor arrays enables an E-Nose with differentiation capability towards different target gases and the differentiation capability of an E-Nose can be further improved with advanced modeling techniques. Table 2 lists the surveyed E-Nose modeling practices for qualitative aroma analysis.

Many studies [5,6] have adopted linear models such as linear discriminant analysis (LDA) or the crude k-nearest neighbor (KNN) model due to their easy implementation. Some studies have experimented with simple neural networks [66,75], most of which consisted of only one or two layers with a small number of parameters. E-Nose data are hard to obtain since there is no standard E-Nose system configuration or setup. In addition, environmental conditions vary among experiments, which holds the E-Nose back from adopting deep learning methods that demand large amounts of data samples.

However, there are also practices to apply deep learning models such as deep multi-layer perceptron networks, LSTM [78], and convolution neural networks (CNN) for odor classification [79,80,81]. It was reported by [82] that a deep neural network (DNN) with five hidden layers outperformed an SVM and MLP with a single but wide hidden layer for classifying wine. The proposed DNN required manual feature extraction and the maximum responses of sensors were used as features.

Regular feed-forward neural network architecture such as MLP tremendously increases the number of parameters when the network goes deeper. However, the convolutional neural network (CNN) improves efficiency by reusing the same set of parameters to different segments in inputs, which is well suitable for applications with inputs highly correlated in local areas, such as images and time-series signals. Compared to traditional machine learning methods, the CNN does not require the feature extraction process.

The architecture of a CNN can vary depending on the interpretation of E-Nose data. Some recent studies considered E-Nose data as a time-series array, while others interpreted E-Nose signals as images and applied a similar CNN architecture to image processing. Zhao et al. [79] processed e-nose signals as time-series data using a 1D-CNN, taking the assumption that signal responses from the gas sensor array in an E-Nose correlated along time steps, and signals from different sensors were independent. A structure was proposed (Figure 6) as a combination of two topologies: (1) signals of each sensor are processed by the same convolution operation and then concatenated along a depth channel, (2) cross sensor relationships are considered in the following three 1D convolution layers. A dropout layer was added during training to avoid overfitting, and a uniformly distributed Xavier was used for convolution layer parameter initialization. A total of 593 samples were used for model training and an evaluation in differentiating Ethylene, CO, and Methane. It was reported that 1D-CNN could outperform SVM, MLP, KNN, and random forest by around 10% on average.

On the other hand, Qi et al. [83] treated the E-Nose signals as an image and used a CNN with two convolution layers for Chinese liquor classification. All the values in the gas sensor array signal map were normalized to between 0 and 1 to generate a grayscale image. Wei et al. [73] tried to adapt a LeNet-5 network structure, which was previously for handwritten letter recognition to classify gases. They first down-sampled the signal to smaller feature maps, then rescaled all the values to 0~255, and then fed these to the CNN network as shown in Figure 7**.** To remedy the lack of data problem, a data augmentation technique was applied through translating the down-sampled data by steps of 2n to have a new feature map. It was reported that the resulting model outperformed multi-layer perceptron (MLP) and other linear models when classifying three gases and gas mixtures.

Peng et al. [84] practiced a deeper CNN with a total of 12 convolution layers and two pooling layers. Inspired by ResNet, shortcuts across convolution blocks were included to overcome the problem of gradient vanishing and speed up training. In their setting, four types of gases were measured with eight MOX gas sensors, with each gas sampled 300 times to form a dataset of 1200 samples. The proposed GasNet structure achieved 95.2% classification accuracy, outperforming an MLP and SVM by a large amount but taking training speed and model size as trade-offs. Zhang et al. [85] added a channel attention module to the CNN backbone, which was used to learn the dependencies of different channels for refined features. Compared to the manual feature extraction, the proposed model resulted in the best classification accuracy on 10 Chinese liquors.
(3)$$Hc_{i,j}=\sqrt{\frac{1+\cos\left(\theta_{i}+\theta_{j}\right)}{2}},\;1\leq i,j\leq T$$
(4)$$Hs_{i,j}=\sqrt{\frac{1-\cos\left(\theta_{i}+\theta_{j}\right)}{2}},\;1\leq i,j\leq T$$

Some recent works encoded E-Nose data into an image before adopting a CNN for classification. Liu et al. [86] encoded the time-series signal of each sensor to an image of three channels to preserve the temporal dependencies. Two out of three total channels in the encoded images were built from a polar transition field, and the other channel was built from a Markov transition matrix. To convert time-series signals to a polar transition matrix, response strength in a sensor signal was normalized by the min–max method to the range of 0~1. Thus, an angle can be calculated as the inverse cosine of the normalized response. Two T × T angle transition matrixes were built with each position as Equations (3) and (4), where T is the number of time steps for the signal. In addition, the Markov transition matrix represented the chance that a state appears at a specific time after a state at another time point. The bin method constructed different states of responses, which quantile the response value to Q slots and assign the sequence number of bins as the state for each response value (Figure 8). The Q value was reported to affect the classification performance and was set to 32 for the optimal result. All the images encoded from nine sensors would be patched together to form an image of dimension (3T) × (3T) × 3 in RGB format for visualization and CNN classification. A recent study from Wang et al. [87] surveyed several ways to convert E-Nose data to an image-like 2D structure, including (1) taking E-Nose time-series data as an image, (2) reshaping data from each sensor into a small image patch and padding all the patches together in order, (3) the same as (2) but putting the most relevant sensors closer during padding. A modified ResNet-based CNN network was proposed to perform classification, and the conversion through method (3) was proved to outperform the other two on classification accuracy. Jong et al. [88] converted the correlation coefficient table of sensor responses to a heat map image; a regular CNN can process the resulting images for image processing. Shi et al. [89] treated the correlation coefficient table as a complete graph and used a graph convolutional neural network for modeling.

Terros-Tello et al. [90] investigated the performance of 1D-CNN, LSTM, and traditional machine learning models on classifying odors from explosives of different amounts. The LSTM model was able to produce accurate prediction by examining only a short portion of the entire time-series of E-Nose data.

To further improve the differentiation capability for gases, sensor fusion has been introduced by some studies by considering the signals from an electronic tongue together with an E-Nose [5,66,72,91], and Figure 9 shows one of the sensor fusion frameworks. Two sets of features are extracted from the E-Nose and E-Tongue, respectively, and are used to train separate classifiers, a decision is made by fusing the result from both classifiers using decision level fusion based on Dempster–Shafer evidence theory.

#### 2.2.2. Quantitative Aroma Analysis

Quantitative analysis of E-Nose signals is a task aimed at estimating the continuous properties associated with gases or odors such as molecule concentration and strength. Compared to qualitative analysis, where labels associated with each sample indicate only the uniqueness of certain gases, target labels/properties in the quantitative analysis are more flexible. For instance, Zhang et al. [15] used a gas sensor array to estimate the concentration of formaldehyde, ammonia, and their mixtures. An MLP model was built in the experiment for prediction, and the mean absolute errors were 0.27 ppm and 0.37 ppm for ammonia and formaldehyde, respectively. In addition to the concentration of specific molecular components, other properties associated with gas can be predicted if they are related to the molecular composition of the gas. Coffee pH may relate to the concentrations of different volatile compounds, and it can be predicted based on the signal generated by a gas sensor array [66]. Table 3 provides a summary of practices for quantitative aroma analysis with an E-Nose.

Linear models are the most common models used in quantitative aroma analysis with an E-Nose for their simplicity. Partial least square regression (PLSR) is the preferred linear model over regular linear regression [70]. The preference is due to the limitation of the E-Nose data: features are much cheaper to calculate than obtaining many data samples, which introduces the curse of dimensionality problems. In this case, a high correlation might be observed among features, and overfitting can occur. PLS decorrelates features by projecting them into a latent space and reducing feature numbers by keeping the top-k latent variables that most explain the variance of latent target variables.

Recently, there has been a rising popularity in applying neural network structure for quantitative aroma analysis. Multi-layer perceptron (MLP) is the simplest feed-forward neural network structure applied by many studies. At each hidden layer, matrix multiplication is performed between the input vector and weight matrix to produce the output at that specific layer Most researchers adopt the architecture with one~two hidden layers and hidden units of different sizes, but networks can vary with structure towards predicting targets. Zhang et al. [92] reported their experiment using an E-Nose to predict individual chemical concentrations in gas mixtures with MLP (Figure 10). They compared the performance between a multiple inputs multiple outputs (SMIMO) MLP and several multiple inputs single output MLPs (MMISO).

Radial basis function network (RBFN) is a type of neural network with special architecture, which normally does not go “deep” (Figure 11). Typical RBFN consists of an input layer, a hidden layer, and an output layer. Unlike MLP, whose parameters are all randomized before training and calculations are performed as a matrix calculation at all the hidden layers, RBFN uses a radial basis function $\phi \left(d\right)$ to calculate the response at each hidden unit, where d represents the distance between the “center” of the unit and input vector. Gaussian activation (Equation (3)) is often used as the radial basis function in RBFN, which plays a similar role as the RBF kernel in a support vector machine to add more non-linearity by virtually projecting input vector into higher dimensions. Moreover, RBFN has a different training method from MLP, which relies mainly on the gradient-based method. The training step varies for RBFN depending on the decision of the variable setting: if center vector and receptive width are to be updated during training, then the gradient-based method can be used; otherwise, training will adopt a two-step manner, the hidden layer and output parameters will be trained separately in each step. To train the hidden layer, a center-based cluster algorithm such as K-means is used to determine the mean and variance of each hidden unit, and these variables are fixed during the training for the output layer using the gradient-based method [95]. RBFN is generally faster and more robust to train than MLP and is likely to perform better than MLP for qualitative prediction.
(5)$$\phi \left(x,c\right)=e^{-\frac{{\left|\left|x-c\right|\right|}^{2}}{2\sigma^{2}}}$$

There have been many recent studies using deep learning models for quantitative aroma analysis with an E-Nose [96]. Wang et al. [97] trained different recurrent neural network models on an open source dataset for air pollutants’ concentration. Compared to the vanilla recurrent neural network (RNN) and gated recurrent unit (GRU), LSTM showed the lowest prediction error on all four different pollutants. Guo et al. [98] proposed an E-Nose framework to predict odor descriptors using a CNN–LSTM model. E-nose data collected from 16 gas sensors were first sliced into small patches, among which each patch represented the same period. The data patches were then fed into multiple CNN–LSTM models, and each of the models ended with a fully connected layer that regressed a combination of odor descriptors. The CNN–LSTM model used both the spatial and temporal information while avoiding the gradient vanish problem in LSTM for long time series. To solve the data contamination problem caused by noise, Wijaya et al. [99] performed noise filtering through wavelet transform on the E-Nose raw signals before feeding the signals into an LSTM model. The most suitable mother wavelet for wavelet decomposition was decided based on the information quality ratio between raw and filtered signals. The noise filtering step was proved to be important to LSTM model performance on predicting the microbial population in beef samples.

Various metrics can be used to evaluate the performance of modeling for a gas property estimation task. Table 4 summarized the description and equation (if any) for evaluation metrics, among which a large t-value, R-value, and R^2^ value are desirable, while a small RMESP, RMSE, and MSE are desirable.

### 2.3. Sensor Drift Compensation

One of the biggest problems existing in gas sensor applications is sensor drifting. There are two causes of drifting: (1) natural drift is due to the aging of the sensor, and (2) secondary drift is due to environmental influences such as temperature and humidity. Unlike other sensors such as the gyroscope or accelerometer, gas sensors require reference gases of specific concentrations for calibration. Romain et al. [101] had a long-term stability test for commonly used MOX gas sensors and found that some sensors drift more than 200% after 7 years. A dataset with gas sensor signals over 36 months for investigating sensor drift was released by Vergara et al. [35], which was used by many related studies as the benchmark for gas sensor drift compensation. The dataset was divided into 10 batches, with all the data samples in the same batch corresponding to specific time ranges. Multiple machine learning techniques addressing the sensor drifting problem have been proposed such as ensemble learning and domain transfer learning. Table 5 summarizes the methods adopted by the recent E-Nose drift compensation based on the classifier ensemble, and all the methods were evaluated on the dataset collected in [35].

Ensemble learning for drift compensation trains a set of predictors on the data collected at different times [106,107]. Each of the predictors is assigned a weight on predicting new data samples. The average accuracy and standard deviation of all data batches and the final data batch accuracy are compared among different methods. Vergara et al. [35] proposed the ensemble learning method with SVM classifiers to counteract sensor drifting. An SVM classifier was trained on each batch of newly collected data at time t, notated as $f_{t}\left(x\right)$. To predict the data at time step *t+1*, the decision was made as a weighted sum of classifiers trained previously, i.e., $h_{t+1}=\sum_{i=1}^{t}\beta_{i}f_{i}\left(x\right)$, where $\;\beta_{t}$ is the weight for each classifier. For simplicity, the prediction accuracy of each $f_{t}\left(x\right)$ on the current batch of data was used as $\beta_{t}$. The result showed that the ensemble classifier improved classification stability (Figure 12). Liu et al. [103] improved the ensemble method by introducing extra weights when training each classifier, called 2D dimension ensemble. For each batch of data with k different classes, *k*(*k* − 1)/2 classifiers were trained to solve the multi-class classification problem. Each of the classifiers was assigned a weight based on their performance on the current batch of data. Verma et al. [36] modified the original optimization step in ensemble drift compensation by introducing a regularization term. The regularization term restricts the data distribution change using the KL divergence and norm-based terms, resulting in higher accuracy than the original approach. Zhao et al. [105] proposed an ensemble learning framework with SVM and LSTM classifiers for drifting compensation. The dataset was preprocessed in four different configurations for training both SVM and LSTM, which added extra robustness to the ensemble learning.

Domain transfer learning aims to find a feature space that maximizes the similarities between samples from the source domain and target domain while ensuring the discrimination capability of the features [108,109,110]. Zhang et al. [111] investigated domain transfer learning with extreme learning machine (ELM) for gas sensor drift compensation. ELM is a special MLP whose weights in the first two layers are randomized and only the weights in the last layer are tunable. Two proposed methods (DAELM-S and DAELM-T) updated the weights parameter β of ELM obtained from previous data (source domain) by incorporating the latest partially labeled and unlabeled data (target domain). DAELM-S (Equation (6)) updated β by minimizing the weighted sum of error for labeled data from the source domain and target domain, where t is the target value and H is the input to the output layer; DAELM-T (Equation (7)) updated β by minimizing the error on the latest labeled data while regularizing the parameters change based on the source domain parameters.
(6)$$\underset{\beta S}{\min}L_{DAELM-S}=min_{\beta S}\frac{1}{2}{∥\beta_{S}∥}^{2}+\frac{C_{S}}{2}\sum_{i=1}^{N_{S}}{∥t_{S}^{i}-H_{S}^{i}\beta_{S}∥}^{2}+\frac{C_{T}}{2}\sum_{j=1}^{N_{T}}{∥t_{T}^{i}-H_{T}^{i}\beta_{S}∥}^{2}$$
(7)$$\underset{\beta T}{\min}L_{WDAELM-T}=min_{\beta T}\frac{1}{2}{∥\beta_{T}∥}^{2}+\frac{C_{T}}{2}\sum_{i=1}^{N_{T}}{∥t_{T}^{i}-H_{T}^{i}\beta_{T}∥}^{2}+\frac{C_{Tu}}{2}\sum_{j=1}^{N_{Tu}}{∥(H_{Tu}^{j}\beta_{B}-H_{Tu}^{j}\beta_{T})∥}^{2}$$
(8)$$\underset{\beta T}{\min}L_{WDAELM-T}=min_{\beta T}\frac{1}{2}{∥\beta_{T}∥}^{2}+\frac{C_{T}}{2}\sum_{i=1}^{N_{T}}{∥t_{T}^{i}-H_{T}^{i}\beta_{T}∥}^{2}+\frac{C_{Tu}}{2}\sum_{j=1}^{N_{Tu}}{∥w\circ \left(H_{Tu}^{j}\beta_{B}-H_{Tu}^{j}\beta_{T}\right)∥}^{2}$$

Based on [111], Ma et al. [112] proposed the weighted domain transfer extreme learning machine (WDTELM), which focused on reducing the impact of wrongly classified unlabeled data samples to parameter update. The final objective function was a variation from DAELM-T with extra weight (Equation (3)). Unlabeled data were first clustered and only a few in each cluster needed labeling. The weight for the rest of the unlabeled sample was assigned by its distance to the labeled data from the same cluster. The method showed 4% improvement on average compared to the previous DAELM-T.

Zhang et al. [113] proposed to find a feature projection that mitigated the difference between the source and target domain while regularizing the data distortion. Similarly, [114] used a kernel transformation for domain transfer. Based on [113], Yi et al. [115] proposed a feature subspace projection method by minimizing local intra-class variance and maximizing local inter-class variance. An autoencoder-based domain transfer was proposed in [34] to learn a feature projection unsupervised. The method modeled sensor drift as a result of device variation and time variation and encoded both variations into a domain feature vector. The autoencoder (Figure 13) was optimized to recover the input feature at its output. For those labeled data in the target domain, the resulting feature representations from the autoencoder were fed into an MLP for further classification. During training, a regularization term was added to enforce the similarity between encoded features for samples from the source and target domains.

Tao et al. [116] developed an adversarial training framework based on neural networks for E-Nose domain adaption, and the Wasserstein distance was used to measure the difference between the source domain and target domain (Figure 14). The adversarial training framework consisted of a feature extractor and a domain discriminator, where the feature extractor was trained to generate similar features for samples from the source and target domain, and the domain discriminator was trained to maximize the dissimilarities between the two. The classification performance was added as another constraint for the extracted features.

In addition to domain transfer learning, Atiq et al. [117] proposed a method to select drift-insensitive features. Discrete binary particle swarm optimization (DBPSO) searched for top-M feature combinations from the feature space that are the most resistant to drift. A cosine similarity model was built to evaluate feature combinations’ drift resistance by training on the first data batch and testing on other data batches collected at different time points. Yu et al. [118] also stated that by using a deep belief network for data preprocessing, the resulting features are more resistant to drifting due to the strengthened coupling among different sensors.

## 3. Conclusions

Due to the complex VOCs composition of odors, machine olfaction is rather challenging. The recent significant improvement in E-Nose’s stability and performance in both qualitative and quantitative analysis is a result of adopting machine learning methods. This review presented an overview of machine learning methods in smart sensing with a focus on feature extraction, modeling, and sensor drift compensation.

Previous works in E-Nose technology have extensively studied the time-domain and frequency-domain features in the signal analysis of an E-Nose. Manually extracted features were sufficient for odor discrimination in many cases and were widely used in various applications. However, manual feature extraction requires prior knowledge of gas sensor technology and needs very careful and time-consuming feature selection. In contrast, recent studies showed successful feature learning of raw sensing signals with neural networks, such as deep belief network and autoencoder, which only needed minimum data preprocessing steps for very competitive odor prediction accuracy. In addition, many practices adopted neural networks for modeling in both qualitative and quantitative analysis with E-Nose. Even with the limited data samples, the reported CNN and LSTM architectures led to performance boosts in comparison with conventional machine learning models. Moreover, gas sensor drifting affects the signal and feature consistency, which is a critical problem for an E-Nose’s performance. Although gas sensor drift compensation was addressed by many recent works with machine learning methods such as ensemble learning and domain adaption learning, it remains a significant obstacle for E-Nose technology. Additionally, the performance of feature selection and models depends on the E-Nose system setup and target gas [119,120,121], which might be another challenge to overcome.

Given that many advanced machine learning techniques are well established for other fields such as audio processing and computer vision [122], we hope that more attempts can be made to migrate those methods to E-Nose applications in the future. In addition, most works reported the proposed machine learning algorithm performance on a dataset collected on their own device, which makes it difficult to compare algorithms across different works. Although there have been a few public E-Nose datasets available [123,124,125], they have a limited number of samples and focus on specific target gases. Therefore, there is a need for building a benchmark dataset for the E-Nose with a standard system setup and data collection scheme. Moreover, many current works addressed the effectiveness of their methods only on certain target gases. However, we envision common patterns are shared among different gases of similar odors and transfer learning across various gases should be further exploited [120].

## Figures and Tables

**Figure 1 sensors-21-07620-f001:**
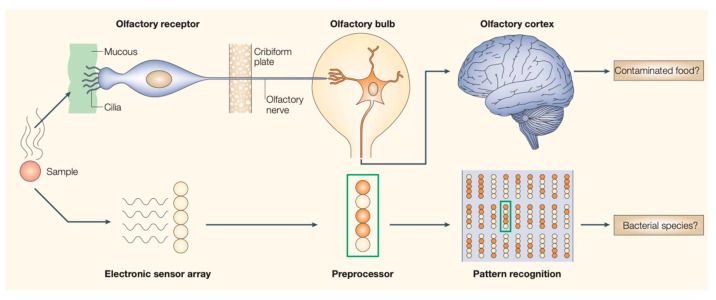
An analogy between human olfactory system and E-nose [33].

**Figure 2 sensors-21-07620-f002:**
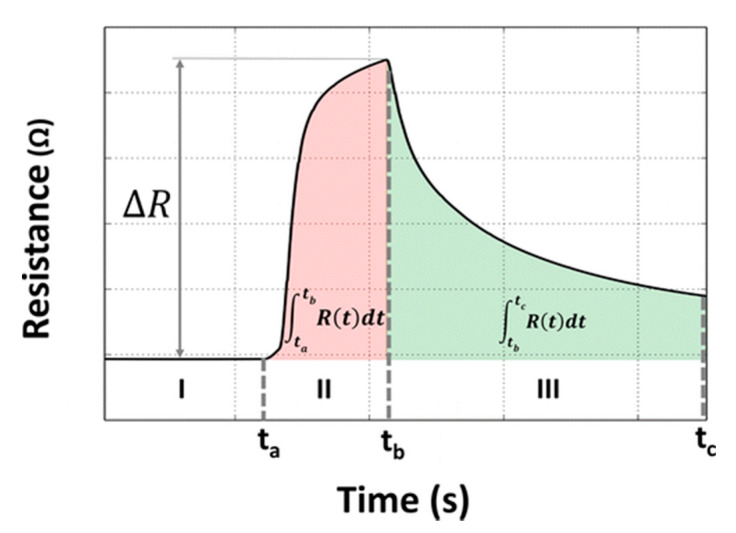
A typical electrical response from a gas sensor [24].

**Figure 3 sensors-21-07620-f003:**
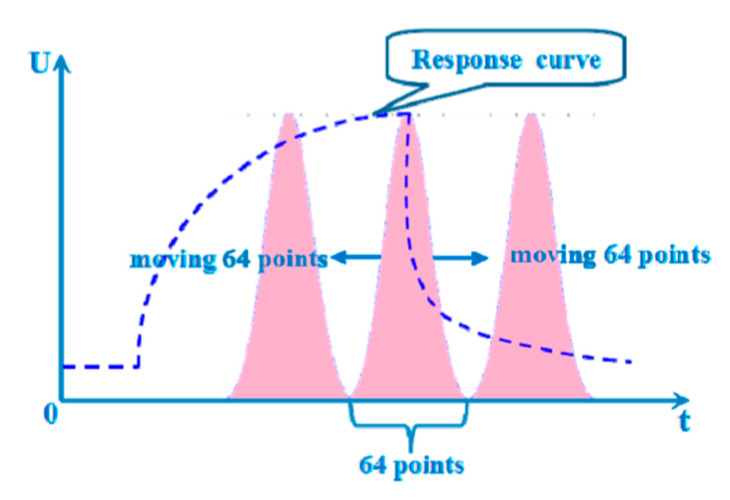
The schematic diagram of moving window function capturing (MWFC) [46].

**Figure 4 sensors-21-07620-f004:**
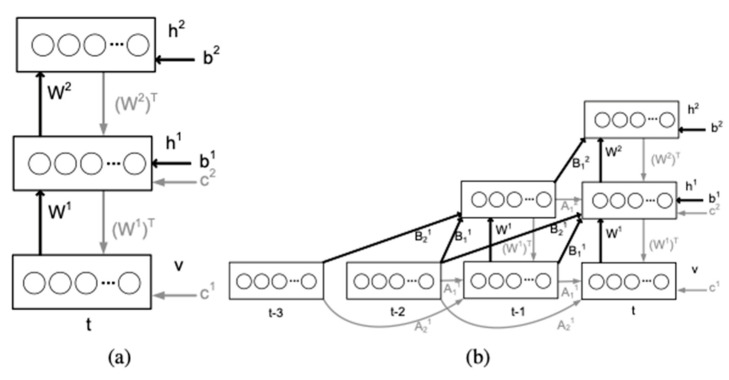
(**a**) Deep RBM and (**b**) deep cRBM model [57].

**Figure 5 sensors-21-07620-f005:**
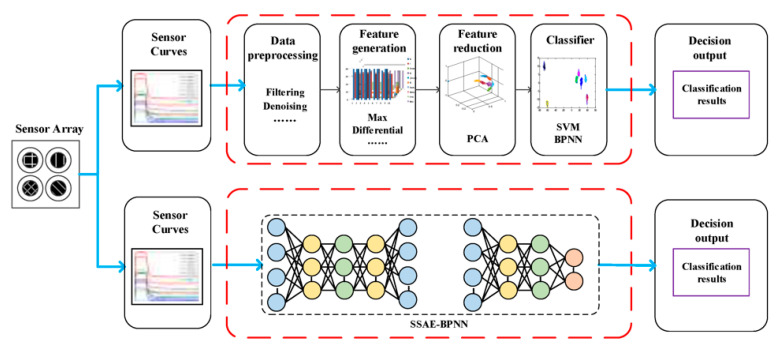
Autoencoder for feature extraction [62].

**Figure 6 sensors-21-07620-f006:**
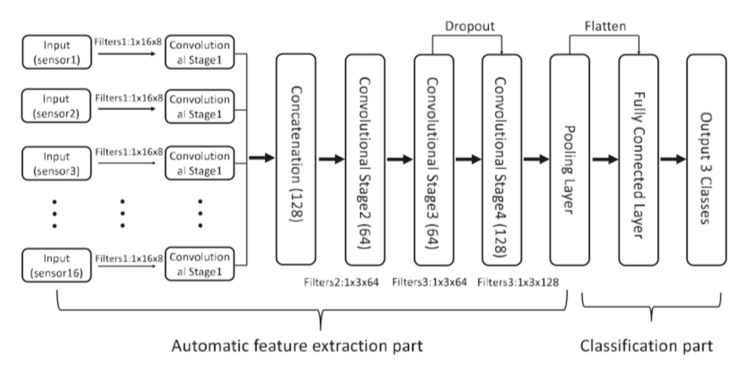
Architecture of 1d-DCNN algorithm [79].

**Figure 7 sensors-21-07620-f007:**
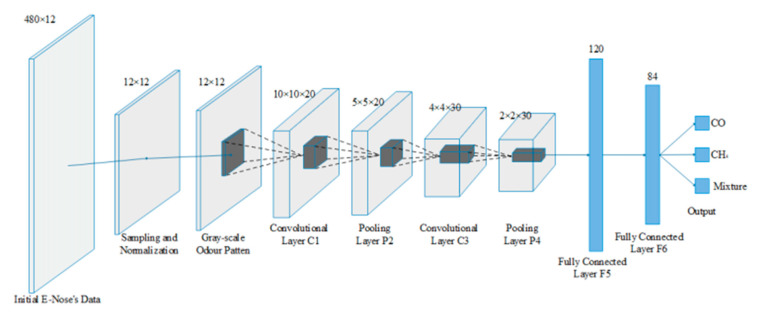
LeNet-5 structure for odor identification [73].

**Figure 8 sensors-21-07620-f008:**
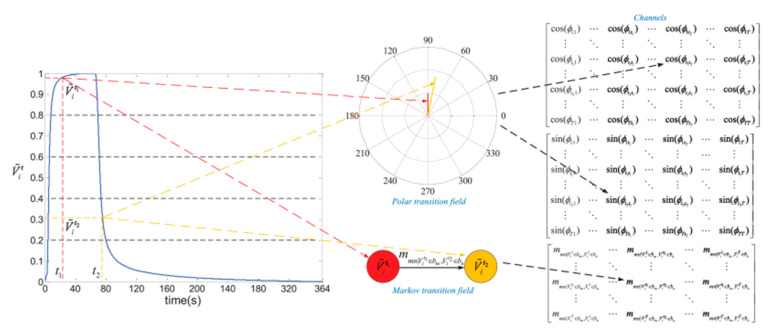
Illustration of encoding sensor response into RGB image [86].

**Figure 9 sensors-21-07620-f009:**
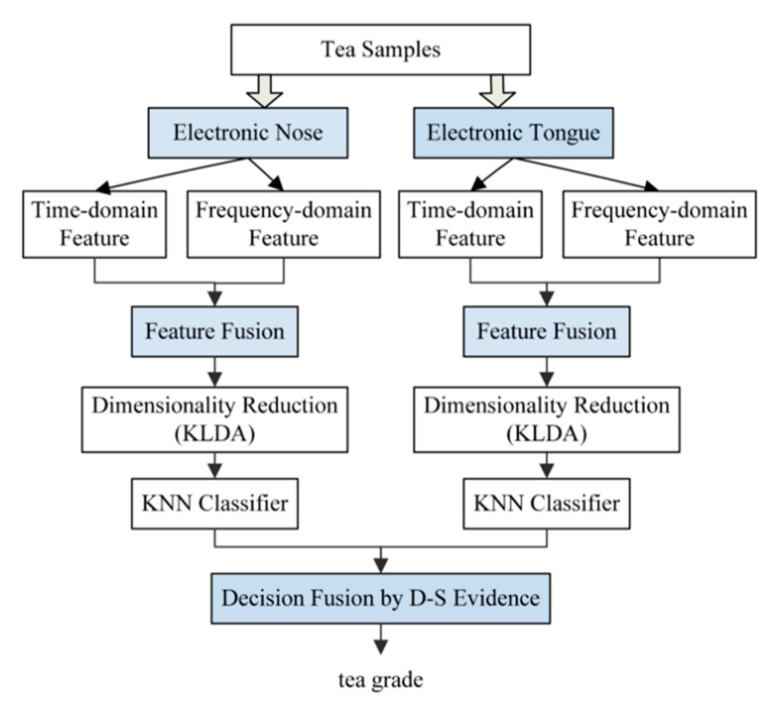
A sensor fusion framework to predict tea of different grade [5].

**Figure 10 sensors-21-07620-f010:**
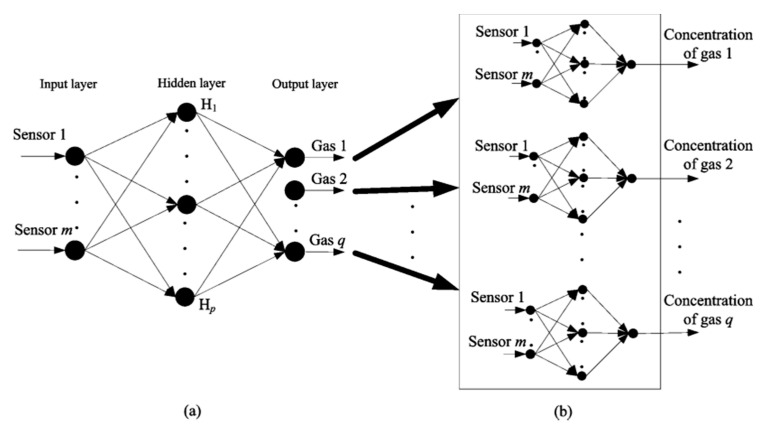
Structure of (**a**) SMIMO- and (**b**) MMISO-based MLP concentration estimation models [92].

**Figure 11 sensors-21-07620-f011:**
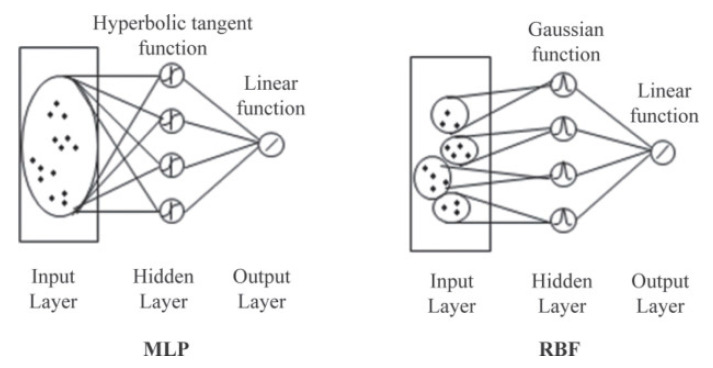
MLP and RBFN [100].

**Figure 12 sensors-21-07620-f012:**
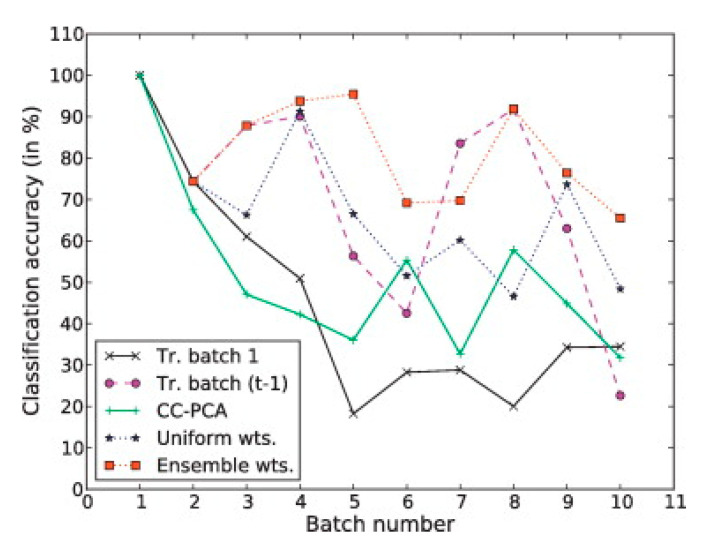
Validation result for different drift compensation methods [35].

**Figure 13 sensors-21-07620-f013:**
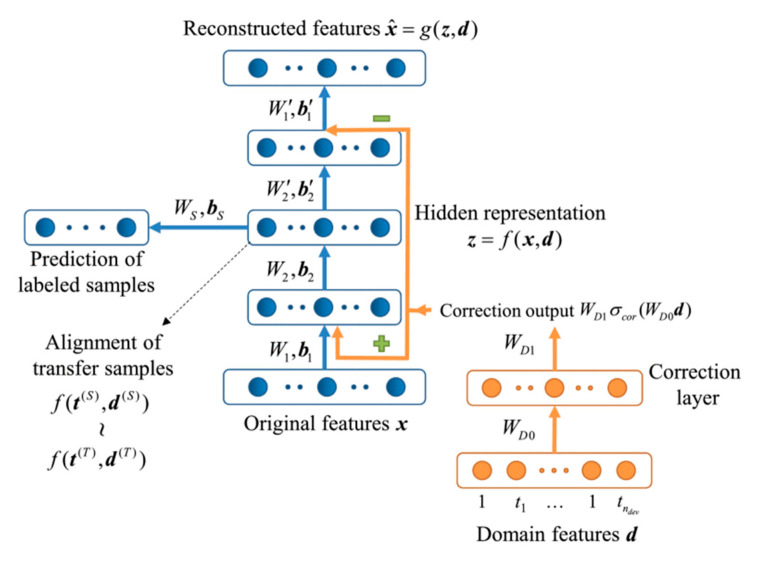
DCAE with correction layer and hidden layers [34].

**Figure 14 sensors-21-07620-f014:**
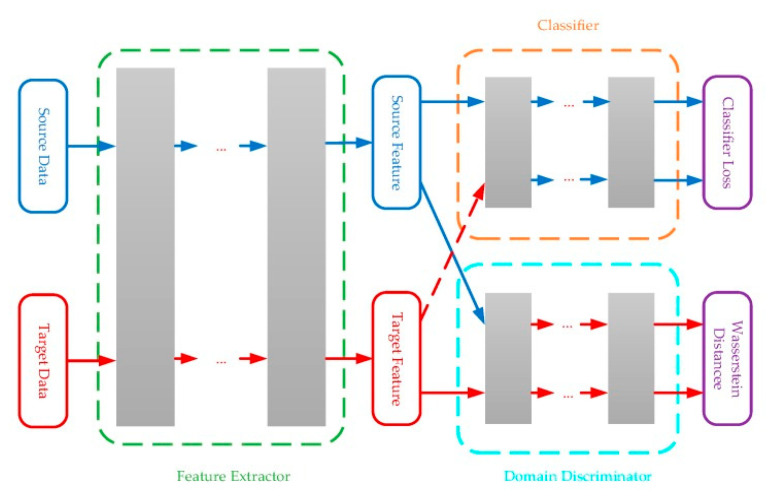
Adversarial training framework for gas sensor drift compensation [116].

**Table 1 sensors-21-07620-t001:** Summary of commonly used features extracted from time-response curves.

Feature	Description
Maximum response	Max value of response
Responses of special time	Response value of special time in the whole response curve
Time of special responses	Time of special response value in the whole response curve
Area	Area values of sensor response curve and time axis surrounded
Integral	Area between two time points
Derivative	$D=\frac{dx\left(t\right)}{dt}$
Difference	Magnitude difference between two time points
Second Derivative	$D^{\prime \prime }=\frac{d^{2}x\left(t\right)}{dt^{2}}$

**Table 2 sensors-21-07620-t002:** Summary of commonly used model in E-Nose odor differentiation.

Gas Type	Gas Number	Sensor Number	Models	Reference
Tea	4	18	KNN, and variations, LDA	[6]
Tea	4	18	KNN ensemble	[5]
Coffee	7	6	KNN, PLS-DA, Multi-Layer Perceptron	[66]
Beer	5	10	SVM	[52]
Beer	5	10	CNN–SVM	[55]
Wine	3	6	SVM, XGBoost, Multi-Layer Perceptron	[67]
Onion	2	7	LDA	[68]
Potato	5	9	LDA, Multi-Layer Perceptron	[69]
Rice	6	10	Extreme Learning Machine	[56]
Ginseng	4	18	LDA, Hierarchical Cluster Analysis	[70]
Kiwifruit	8	10	LDA	[71]
Soy Sauce	4	18	-	[72]
Single Chemicals	20	1	KNN, LDA	[24]
Single Chemicals	4	1	KNN, LDA, Random Forest	[23]
Single Chemicals	3	1	SVM	[53]
Single Chemicals	3	12	CNN	[73]
Single Chemicals	12	8	CNN	[74]
Polluted water	12	4	Multi-Layer Perceptron	[12]
Smell mixture	10	7	Multi-Layer Perceptron	[75]
Essential Oils	6	9	Multi-Layer Perceptron	[76]
Essential Oils	6	9	LDA, SVM	[77]

**Table 3 sensors-21-07620-t003:** Summary of commonly used model in E-Nose gas concentration estimation.

Gas Type	Predicting Property	Predicting Target	Sensor Number	Models	Evaluation Method	Reference
Gas Mixture	Concentration	3	2	Neural-fuzzy network	RMSE	[14]
Gas Mixture	Concentration	6	4	MLP and its variations	MSEP	[92]
Gas Mixture	Concentration	2	3	MLP	MSE, MAE	[15]
Ginseng	Chemical Concentration	7	18	PLSR, MLP	RMSE, R^2^	[35]
Tea	Chemical Concentration	4	10	SVMR, Random Forest Regression	RMSE, R^2^	[7]
Fish	TVC	1	9	SVMR, RBFN	RMSEP, R-value	[91]
Flower	Aroma Strength	1	11	MLP, RBFN	RMSE, R^2^	[93]
Coffee	PH, Solid%. Acid%, Soluble%	4	6	PLSR	R-value, RPD, RMSE	[66]
Beer	Chemical Concentration	-	9	MLP	R-value, MSE	[94]
Squid	Chemical Concentration	1	18	PLSR	R^2^, *t*-test	[8]
Polluted water	Odor Concentration	1	5	PLSR	RSME, R^2^	[13]
Kiwifruit	Ripeness Index	3	10	PLSR, SVMR, Random Forest Regression	RSME, R^2^	[71]

**Table 4 sensors-21-07620-t004:** Modeling evaluation metric for gas property estimation.

Metric	Equation	Description
t-value	-	The significance of the predicting model is close to the real model
r-value	$\frac{\sum_{i=1}^{n}\left(\hat{y_{i}}-\bar{\hat{y}}\right)\left(y_{i}-\bar{y}\right)}{\sqrt{\sum_{i=1}^{n}{\left(\hat{y_{i}}-\bar{\hat{y}}\right)}^{2}\sqrt{\sum_{i=1}^{n}{\left(y_{i}-\bar{y}\right)}^{2}}}}$	The correlation between predicted value and real value
R^2^	$1-\frac{\sum_{i=1}^{n}{\left(\hat{y_{i}}-y_{i}\right)}^{2}}{\sum_{i=1}^{n}{\left(y_{i}-\bar{y}\right)}^{2}}$	The extent to which predict model is explaining the variation of data
RMESP	$\sqrt{\frac{\sum_{i=1}^{n}{\left(\frac{\hat{y_{i}}-y_{i}}{y_{i}}\right)}^{2}}{n}}$	Average squared rooted deviation from predicted value to real value by percentage
RMSE	$\sqrt{\frac{\sum_{i=1}^{n}{\left(\hat{y_{i}}-y_{i}\right)}^{2}}{n}}$	Average squared rooted deviation from predicted value to real value
MSE	$\frac{\sum_{i=1}^{n}{\left(\hat{y_{i}}-y_{i}\right)}^{2}}{n}$	Average squared deviation from predicted value to real value
MAE	$\frac{\sum_{i=1}^{n}\left|\hat{y_{i}}-y_{i}\right|}{n}$	Average absolute deviation from predicted value to real value

**Table 5 sensors-21-07620-t005:** Summary for drift compensation by ensemble.

Ensemble Method	Accuracy Mean	Accuracy Std Dev	Accuracy on Final Batch	Reference
SVM	~81.6%	~12%	~68%	[35]
MLP and KNN	63.93% (MLP), 75.59% (KNN)_	~29% (MLP), ~17%(KNN)_	38% (MLP), 53% (KNN)	[102]
SVM with 2D weights	84.8%	~15%	~60%	[103]
SVM with regularization	~79.3%	~8%	~80%	[36]
MLP	~83.1%	~10%	72.89%	[104]
SVM, LSTM	83.2% (SVM) 77.8% (LSTM) 89.26% (SVM and LSTM)	16.63% (SVM) 9.21% (LSTM) 10.0% (SVM and LSTM)	70.6% (SVM), 83.3% (LSTM), 83.4% (SVM and LSTM)	[105]

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
