# Peer review of "Recent Progress in Smart Electronic Nose Technologies Enabled with Machine Learning Methods"

_sensors, 2021, doi:10.3390/s21227620_

Round 1
Reviewer 1 Report
The subject presented in the manuscript “Recent Progress in Smart Electronic Nose Technologies Enabled with Machine Learning Methods” is interesting and the review is suitable to be published in Sensors. Therefore, I recommend this manuscript for publication after minor revision, and some general comments/suggestions/questions are in the following:
1) Please check the affiliation details, as they seem incomplete.
2) Although there are many review articles on e-nose and machine learning in the literature, there is still room to review the recent advances in this area. Having said that, how would the authors compare their manuscript with the other reviews already published on the same subject? It may be interesting to mention similar reviews in the introduction section (or wherever you find suitable), pointing their strength and what lacks in the literature, then highlighting what is outstanding in your manuscript.
3) Section 2.1, line 2: There is an error in the cross-reference citation (probably some inconsistency in your reference management tool). The same error is repeated throughout the manuscript.
4) Table 1: Is x(t) the same as R(t) of Figure 2? Please clarify.
5) Figure 3: The acronym MWFC was not defined in the main text.
6) Page 4, paragraph 4: The authors mention that the frequency-domain features are extracted by transforming the original time-domain signal. I would like to highlight that frequency-domain response may be also obtained directly from impedance spectroscopy measurements. There are many papers of e-tongues based on impedance spectroscopy measurements, and also some e-nose based on this technique can be found.
7) Figure 5: The acronyms SSAE and BPNN were not defined in the main text.
8) Page 7, line 2: Please define LSTM.
9) Table 2: Some acronyms such as KNN, HCA, and LDA were not defined in the main text. Please check the other acronyms as well.
10) Page 8, paragraph 3: The authors mentioned “figure 2”, but I think the cross-reference is misplaced.
11) Page 10, paragraph 2: Please define “D-S evidence”.
12) Table 5 seems a bit poor. Please add some other details to compare the cited papers.
13) Beyond describing the state of the art of a subject, a review must highlight what lacks in the area to assist and guide other researchers in the field. In my opinion, the authors may improve this manuscript by adding some of their thoughts on this matter in the conclusion section.
Reviewer 2 Report
The title is: Recent Progress in Smart Electronic Nose Technologies Enabled with Machine Learning Methods.
The aim of this study was review the recent advances in E-Nose machine learning techniques, with a focus on three important aspects: 1) feature extraction, 2) modeling, and 3) gas sensor drift compensation. By surveying machine learning methods for different E-Nose applications, this work tried to evaluate the performance of E-Nose in existing applications and inspire new applications in many emerging fields.
I commented on the manuscript and the comments are presented below:
The Introduction of the study is too brief and provides with some general information about the aroma techniques analysis. On the other hand for several decades, studies on application of different type of techniques for detection of odor have been conducted. The same goes for chapter 2. The most part the Machine Learning for E-Nose section is well structured and the obtained data were subjected for statistical analysis. The results were not fully discussed. A full discussion of the results obtained with other work in this field should be carried out in more aspects. the Results section is well structured. The results were not fully discussed. A full discussion of the results obtained with other work in this field should be carried out in more aspects. The authors should supplement the discussion based on publications issued in recent years. I suggest supplementing the Chapters with additional information related to other new methods and devices in research of VOCs detections. „ Identification of the Olfactory Profile of Rapeseed Oil as a Function of Heating Time and Ratio of Volume and Surface Area of Contact with Oxygen Using an Electronic Nose”; „Characterization of fatty acid, amino acid and volatile compound compositions and bioactive components of seven coffee (Coffea robusta) cultivars grown in Hainan Province, China”; “Classification and identification of essential oils from herbs and fruits based on a mos electronic-nose technology”; “Opto-electronic nose coupled to a silicon micro pre-concentrator device for selective sensing of flavored waters”, “Performance analysis of mau-9 electronic-nose mos sensor array components and ann classification methods for discrimination of herb and fruit essential oils”, „Coupling MOS sensors and gas chromatography to interpret the sensor responses to complex food aroma: Application to virgin olive oil”, “A Machine Learning Method for Classification and Identification of Potato Cultivars Based on the Reaction of MOS Type Sensor-Array”.
Additional information contained in the Introduction chapter will make the aim of the study will clearly stated.
Part: Conclusion
Conclusions are synthetically described and result from the conducted review but were not fully discussed in the manuscript.
Part: References.
The literature used is appropriate however the authors should supplement this chapter with newer research papers. The cited literature is in an incompatible format than required by the editor. This should be patched to the correct format.
Round 2
Reviewer 2 Report
The authors referred to the comments from the previous review for the manuscript titled: Recent Progress in Smart Electronic Nose Technologies Enabled with Machine Learning Methods. I accept explanations. They supplemented the discussion with a new literature data strengthens the message and importance of information in the manuscript.